# Functional and Anatomical Outcomes of Anti-Vascular Endothelial Growth Factor Treatment for Exudative Age-Related Macular Degeneration with or without Obstructive Sleep Apnea

**DOI:** 10.3390/ijms24087285

**Published:** 2023-04-14

**Authors:** Nan-Ni Chen, Chau-Yin Chen, Jin-Jhe Wang, Heng-Chiao Huang, Wei-Dar Chen, Ching-Lung Chen, Yao-Hsu Yang, Meng-Hung Lin, Ting-Yu Kuo, Chien-Hsiung Lai

**Affiliations:** 1Department of Ophthalmology, Chang Gung Memorial Hospital, Chiayi 61363, Taiwan; 2College of Medicine, Chang Gung University, Taoyuan 33302, Taiwan; 3Department of Optometry, Chung Hwa University of Medical Technology, Tainan 71703, Taiwan; 4Department of Traditional Chinese Medicine, Chang Gung Memorial Hospital, Chiayi 61363, Taiwan; 5Health Information and Epidemiology Laboratory, Chang Gung Memorial Hospital, Chiayi Branch, Chiayi 61363, Taiwan; 6Department of Nursing, Chang Gung University of Science and Technology, Chiayi 61363, Taiwan

**Keywords:** obstructive sleep apnea, age-related macular degeneration, anti-vascular endothelial growth factor treatment

## Abstract

(1) To investigate the functional and anatomical outcomes of anti-vascular endothelial growth factor (anti-VEGF) treatment in patients with exudative age-related macular degeneration (AMD) with or without obstructive sleep apnea (OSA); (2) In total, 65 patients with AMD with or without OSA who received three consecutive doses of intravitreal anti-VEGF injections were enrolled. The primary outcomes—best-corrected visual acuity (BCVA) and central macular thickness (CMT)—were assessed at 1 and 3 months. Moreover, morphological changes observed through optical coherence tomography were analyzed; (3) In total, 15 of the 65 patients had OSA and were included in the OSA group; the remaining 50 patients were included in the non-OSA (control) group. At 1 and 3 months after treatment, BCVA and CMT had improved but did not differ significantly between the groups. More patients in the OSA group demonstrated subretinal fluid (SRF) resorption at 3 months after treatment than in the non-OSA group (*p* = 0.009). Changes in other imaging biomarkers, such as intraretinal cysts, retinal pigment epithelium detachment, hyperreflective dots, and ellipsoid zone disruptions, did not differ significantly between the groups; (4) Our results suggest that the BCVA and CMT outcomes 3 months after anti-VEGF treatment are similar between patients with and without OSA. Moreover, patients with OSA may exhibit superior SRF resorption. A large-scale prospective study is mandatory to evaluate the association between SRF resorption and visual outcomes in AMD patients with OSA.

## 1. Introduction

Obstructive sleep apnea (OSA) is the most common form of sleep-disordered breathing and is characterized by frequent episodes of obstructed breathing during sleep, resulting in sleep-related decreases (hypopnea) or pauses (apnea) in respiration [1]. The prevalence of OSA is increasing, and it is considered a systemic disease because it is an independent risk factor for metabolic and cardiovascular disorders [2,3,4]. In addition, OSA has been implicated in several ocular diseases, including floppy eyelid syndrome, nonarteritic anterior ischemic optic neuropathy, central serous retinopathy, retinal vein occlusion, and glaucoma [5,6,7]. Accumulating evidence has indicated that patients with OSA are likely to develop age-related macular degeneration (AMD) [8,9]. However, little research has investigated the outcomes after anti-VEGF treatment in AMD patients with OSA [10,11,12]. OSA-related recurrent hypoxia results in increased oxidative stress and inflammation, which are strongly implicated in AMD pathogenesis [13]. Moreover, vascular alterations may be associated with AMD pathogenesis. Decreased vascular flow in the retina and choroid has been associated with AMD development and progression [14,15]. 

It has been reported that patients with exudative AMD and OSA have exhibited poor outcomes after anti-vascular endothelial growth factor (anti-VEGF) treatment [11,12]. In previous investigations, Schaal S et al. found that untreated OSA hinders the response of exudative AMD to intravitreal bevacizumab, and treatment of OSA with continuous positive airway pressure (CPAP) therapy yields a subsequent anatomical response and functional improvement while requiring significantly fewer injections [11]. On the other hand, Nesmith BL et al. found that patients with exudative AMD with poor response to anti-vascular endothelial growth factor therapy have a significantly higher risk of OSA [12]. However, a general consensus has not been reached due to the lack of published RCTs and the small sample size. Furthermore, the effects of anti-VEGF treatment on Asian patients with exudative AMD and OSA remain unclear.

Optical coherence tomography (OCT), which can distinguish fluid in different retinal layers, has been applied in clinical practice to monitor treatment response and may be used to realize the pathogenic vasculature implicated in the disease mechanism and impact of OSA on AMD patients. Therefore, this study employed OCT to investigate the functional and anatomical outcomes of patients with exudative AMD, with or without OSA, after anti-VEGF treatment in Taiwan to gain further insights into the impact of OSA on patients with AMD.

## 2. Results

The baseline characteristics of the study population are presented in Table 1. The study included 65 eyes of 65 patients with a mean age of 71 years. Fifteen of the 65 patients had OSA and were included in the OSA group, and the remaining 50 patients were included in the non-OSA (control) group. The proportion of anti-VEGF used was 21.54% bevacizumab, 40.00% aflibercept, and 38.46% ranibizumab. Among 15 patients with OSA, 9 of them were classified as mild OSA, 4 of them were classified as moderate OSA, and 2 of them were classified as severe OSA based on the American Academy of Sleep Medicine (AASM) classification of OSA severity according to AHI. For moderate and severe OSA, only 2 of them received CPAP, and none of them received surgical treatments. The systemic comorbidities at baseline did not differ significantly between the groups, except for CAD and HTN, which were more prevalent in the OSA group (*p* ≤ 0.001). 

Functional and anatomical outcomes as well as the details of the OCT imaging biomarkers at baseline and 1 and 3 months after anti-VEGF treatment are presented in Table 2. BCVA and CMT at baseline did not differ significantly between the two groups, but choroidal thickness was significantly lower in the OSA group (*p* = 0.003). The morphological patterns—including the grades for IRC, SRF, RPED, HRDs (inner/outer), and SHRM—of the OCT scans were not significantly different between the groups, except for EZ disruption, which was more prevalent (*p* = 0.004) at baseline in the non-OSA group. At 1 and 3 months after treatment, BCVA did not differ significantly between the groups. CMT was lower in the non-OSA group than in the OSA group at 1 month (*p* = 0.030) but was not significantly different between the groups at 3 months (*p* = 0.431). Choroidal thickness was significantly lower in the OSA group than in the non-OSA group at 1 and 3 months after treatment (*p* < 0.001 and *p* = 0.002 at 1 and 3 months, respectively).

Functional and anatomical outcomes after anti-VEGF treatment are summarized in Table 3 and Figure 1. Changes in BCVA, CMT, and choroidal thickness at 1 and 3 months after treatment did not differ significantly between the groups. Changes in SHRM deposition were significantly different between groups at 1 month (*p* = 0.013), with more patients in the OSA group having increased deposition. However, this phenomenon did not persist at 3 months (*p* = 0.785). More patients in the OSA group demonstrated SRF resorption at 3 months than in the non-OSA group (*p* = 0.009). Changes in other imaging biomarkers, including IRC, RPED, HRD (inner/outer), and EZ disruption grades, did not differ significantly between the groups. 

Finally, we divided the OSA group into two categories based on their severity: mild OSA and moderate to severe OSA. Either baseline, 1 month, and 3 months CMT or changes in CMT at 1 month and 3 months from baseline showed no significant difference between the two groups (*p* > 0.05). Moreover, since only two patients received CPAP treatment, we could not further analyze the subgroup due to the small sample size.

## 3. Discussion

OSA is associated with several ocular diseases, such as nonarteritic anterior ischemic optic neuropathy, central serous chorioretinopathy, retinal vein occlusion, glaucoma, diabetic retinopathy, and AMD [6,7,8,9,16]. A possible pathological mechanism underlying AMD involves recurrent hypoxia caused by OSA, which induces angiogenesis for compensatory oxygen provision [17]. Changes in humoral mediators, such as the upregulation of VEGFs or an increase in intravascular hydrostatic pressure, play significant roles in exudative AMD pathogenesis [18]. 

Histological evidence regarding choroidal thickness in AMD is conflicting, with some studies indicating a decrease, particularly in the advanced stages of the disease, and others noting no significant changes, even in advanced AMD [19,20,21,22,23]. This discrepancy can be partially explained by confounding factors such as choroidal hypervascularity, diurnal fluctuation, choroidal thickness, age, axial length, and refractive error. In the present study, choroidal thickness was significantly lower in the OSA group than it was in the non-OSA group at baseline and 1 and 3 months after treatment (*p* = 0.003, *p* < 0.001, and *p* = 0.002 at baseline and 1 and 3 months, respectively). These results are consistent with those of a recent meta-analysis (which included diseases other than AMD), which suggested that choroidal thickness decreased in patients with OSA [19,24]. Thus, intermittent hypoxia and decreased choroidal thickness in OSA may alter retinal and choroidal blood supply, and such a reduced blood flow could aggravate retinal hypoxia [15,19]. 

Notably, BCVA and CMT at 3 months did not differ significantly between the OSA and non-OSA groups. Both groups exhibited significantly decreased CMT and improved BCVA. Furthermore, SRF was resorbed in all eyes in the OSA group at 3 months; this result was significantly different from that of the non-OSA group (*p* < 0.001). Two plausible explanations for these findings are proposed. First, chronic recurrent hypoxia and hypercapnia destroy the vascular endothelium, alter vascular permeability, and subsequently decrease choroidal thickness, possibly leading to progressive dysfunction of the RPE [25]. SRF or macular edema may develop in the early stage of AMD in patients with OSA due to impaired RPE function. Another finding supporting the development of these symptoms in the early stage of AMD in the OSA group was the lower prevalence of EZ disruption at baseline (*p* = 0.004) compared with the non-OSA group. These results may explain the functional and anatomical outcomes of the groups being similar, despite a previous report that OSA may worsen the effect of AMD treatment [12]. Second, hypoxia stimulates the activity of hypoxia-inducible factors and increases the levels of VEGFs, leading to choroidal neovascularization [25,26]. Increased VEGF levels in the retina are responsible for fluid development in AMD with OSA. By contrast, several pathological mechanisms underlying AMD development in patients without OSA, such as changes in the extracellular matrix, changes in Bruch’s membrane composition and permeability, or oxidative mitochondrial damage due to aging or genetic factors, may be partially responsible for their inferior fluid resorption after anti-VEGF treatment [27]. 

Some researchers have suggested that OSA treatments, such as continuous positive airway pressure and upper airway surgery, result in functional and anatomical improvements and significantly reduce the number of necessary IVIs for patients with AMD [11,28]. In Taiwan, OSA is often underdiagnosed; furthermore, the present study applied strict criteria for inclusion in the OSA group. Thus, the patients with OSA in this study who were willing to undergo curative treatment might have been generally health-conscious with early detection of visual change and thereby more likely to achieve favorable visual outcomes after anti-VEGF treatment.

To our knowledge, this is the first study to investigate the differences in changes in OCT biomarkers between patients with AMD, with and without OSA, after anti-VEGF treatment. However, this study has some limitations. First, this was a nonrandomized study, and the data were collected retrospectively from few patients. Second, the follow-up period was short. In a previous study, patients with lower choroidal thickness at baseline appeared to be at higher risk of developing macular atrophy within 18 months [29]. Long-term follow-up for assessing the functional and anatomical outcomes of patients with OSA is necessary because of the severe choroidal thinning in such patients.

In summary, OSA-related recurrent hypoxia may cause changes in angiogenic factors, and patients with AMD and OSA have a lower subfoveal choroidal thickness on OCT. BCVA and CMT outcomes are similar between patients with and without OSA 3 months after anti-VEGF treatment; however, the SRF resorption of patients with OSA is superior. A large-scale prospective study is mandatory to evaluate the association between SRF resorption and visual outcomes in AMD patients with OSA. 

## 4. Materials and Methods

### 4.1. Data Source and Ethics Declaration

We retrospectively reviewed the medical records of patients from Chang Gung Memorial Hospital. The Chang Gung Memorial Hospital Institutional Review Board (IRB: 202000890B0C601) approved this study and, because of the retrospective nature of the study, waived the requirement of patient consent to review medical records. In addition, we anonymized the patient data and maintained confidentiality. We conducted this study in compliance with the principles of the Declaration of Helsinki.

### 4.2. Patient Selection and Study Design

We retrospectively reviewed the Chang Gung research database and retrieved data from January 2006 to April 2020 on patients with AMD [International Classification of Diseases, Ninth Revision, Clinical Modification (ICD-9-CM) and International Classification of Diseases, Tenth Revision, Clinical Modification (ICD-10-CM) codes 362.50, 362.51, 362.52, or H35.32] who received intravitreal injections (IVIs) of anti-VEGF drugs at Chang Gung Memorial Hospital, Chiayi (n = 937). We excluded patients with diagnoses of retinal vein or artery occlusion, those aged <50 years who received IVIs for the first time, those who did not receive three consecutive anti-VEGF doses, those who had missing data or poor-quality OCT images at baseline or at the designated time points, and those who underwent other intraocular surgeries (n = 872). Finally, 65 patients with AMD with or without OSA were enrolled. Anti-VEGF drugs used in our study were bevacizumab, aflibercept, and ranibizumab. For OSA diagnosis, the patients were required to have appropriate ICD-9-CM or ICD-10-CM codes (i.e., 780.51, 780.53, 780.57, G47.0, G47.1, or G47.3) and be admitted for polysomnography.

### 4.3. Baseline Characteristics and Comorbidities

Two unmasked researchers (JJW and NNC) qualitatively and quantitatively graded OCT biomarkers according to a recently published AMD grading protocol [30], which included grades for intraretinal cysts (IRC), subretinal fluid (SRF), ellipsoid zone (EZ) status, subretinal hyperreflective material (SHRM), hyperreflective dots (HRDs), and retinal pigment epithelium detachment (RPED). When disagreement occurred, a third senior retinal specialist would determine the final grading (CHL). Snellen visual acuity measurements were converted to the logarithm of the minimum angle of resolution (log MAR) units for statistical analysis.

The systemic comorbidities considered included arrhythmia, coronary artery disease (CAD), chronic obstructive pulmonary disease, diabetes mellitus (DM), dementia, depression, heart failure, hypertension (HTN), hyperlipidemia, stroke, and obesity. Patients were considered to have one of these comorbidities if they had visited the outpatient or emergency department at least three times within 1 year or were admitted to the hospital at least once before AMD diagnosis. 

### 4.4. Statistical Analysis

The baseline characteristics of patients with and without OSA were compared. Categorical variables are expressed as frequencies and percentages and were compared using the χ2 test. Continuous variables are expressed as medians (interquartile ranges) and were compared using the Mann–Whitney U test. All statistical analyses were performed using SAS, version 9.4 (SAS Institute, Cary, NC, USA).

## Figures and Tables

**Figure 1 ijms-24-07285-f001:**
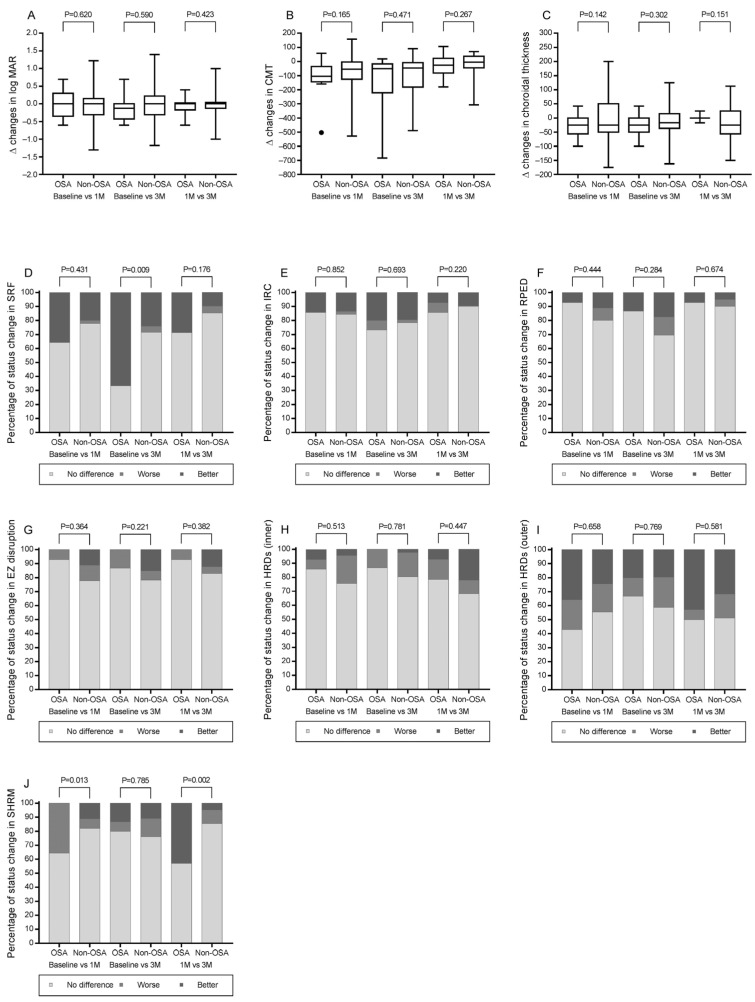
Changes in functional and anatomical outcomes after anti-VEGF treatment. The box plots show differences (medians) between the OSA and non-OSA groups in changes in (**A**) log MAR, (**B**) CMT, and (**C**) choroidal thickness determined from Mann–Whitney U tests. The bar graphs show differences (percentages) between the OSA and non-OSA groups in their changes in (**D**) SRF, (**E**) IRC, (**F**) RPED, (**G**) EZ disruption, (**H**) HRDs (inner), (**I**) HRDs (outer), and (**J**) SHRM from baseline to 1 and 3 months and from 1 to 3 months as determined from χ2 tests. The dot was represented as outlier in subfigure (**B**). Abbreviations: CMT, central macular thickness; EZ, ellipsoid zone; HRDs, hyperreflective dots; IRC, intraretinal cysts; MAR, the minimum angle of resolution; RPED, retinal pigment epithelium detachment; SHRM, subretinal hyperreflective material; and SRF, subretinal fluid.

**Table 1 ijms-24-07285-t001:** Baseline characteristics of study patients.

	OSA		Non-OSA		
Variables	N	(%)	N	(%)	*p*-Value
Gender					0.889
Male	9	(60.0)	31	(62.0)	
Female	6	(40.0)	19	(38.0)	
AMD age, years					1.000
50~65	3	(20.0)	10	(20.0)	
≥65	12	(80.0)	40	(80.0)	
mean (SD)	71.8	(7.5)	71.3	(6.8)	0.824
Comorbidities					
Arrhythmia	4	(26.7)	3	(6.0)	0.024
CAD	7	(46.7)	3	(6.0)	0.001
COPD	1	(6.7)	2	(4.0)	0.551
DM	5	(33.3)	12	(24.0)	0.512
Dementia	1	(6.7)	0	(0.0)	0.231
Heart failure	2	(13.3)	2	(4.0)	0.226
HTN	12	(80.0)	13	(26.0)	0.001
Hyperlipidemia	5	(33.3)	7	(14.0)	0.128
Stroke	3	(20.0)	8	(16.0)	0.706
Obesity	1	(6.7)	0	(0.0)	0.231

Abbreviations: CAD, coronary artery disease; COPD, chronic obstructive pulmonary disease; DM, diabetes mellitus; and HTN, hypertension.

**Table 2 ijms-24-07285-t002:** Functional and anatomical outcomes at baseline and 1 and 3 months after anti-VEGF treatment.

Variables	Eyes (OSA/Non-OSA)	OSA	Non-OSA	*p*-Value
Baseline				
Log MAR	15/50	1.0 (0.7, 1.3)	1.0 (0.7, 1.3)	0.582
CMT (μm)	15/50	416.0 (326.0, 452.0)	342.5 (284.0, 425.0)	0.173
Presence of SRF	15/50	10 (66.7)	37 (74.0)	0.743
Presence of IRC	15/50	6 (40.0)	27 (54.0)	0.342
Presence of RPED	15/50	7 (46.7)	31 (62.0)	0.291
EZ disruption	15/50	5 (33.3)	37 (74.0)	0.004
Presence of HRDs (inner)	15/50	1 (6.7)	9 (18.0)	0.431
Presence of HRDs (outer)	15/50	9 (60.0)	21 (42.0)	0.220
Presence of SHRM (μm)	15/50			0.145
0		9 (60.0)	21 (42.0)	
<500		5 (33.3)	13 (26.0)	
>500		1 (6.7)	16 (32.0)	
Choroidal thickness (μm)	15/50	233.0 (150.0, 275.0)	300.0 (250.0, 350.0)	0.003
1 month				
Log MAR	14/45	0.9 (0.7, 1.3)	0.8 (0.5, 1.4)	0.907
CMT (μm)	14/45	313.0 (285.0, 359.0)	265.0 (224.0, 314.0)	0.030
Presence of SRF	14/45	4 (28.6)	26 (57.8)	0.056
Presence of IRC	14/45	4 (28.6)	18 (40.0)	0.440
Presence of RPED	14/45	6 (42.9)	27 (60.0)	0.259
EZ disruption	14/45	6 (42.9)	32 (71.1)	0.065
Presence of HRDs (inner)	14/45	2 (14.3)	11 (24.4)	0.713
Presence of HRDs (outer)	14/45	8 (57.1)	20 (44.4)	0.542
Presence of SHRM (μm)	14/45			0.831
0		5 (35.7)	20 (44.4)	
<500		5 (35.7)	13 (28.9)	
>500		4 (28.6)	12 (26.7)	
Choroidal thickness (μm)	14/45	191.5 (100.0, 250.0)	287.0 (250.0, 375.0)	<0.001
3 months				
Log MAR	15/46	0.7 (0.4, 1.3)	0.9 (0.5, 1.3)	0.409
CMT (μm)	15/46	276.0 (231.0, 326.0)	268.0 (231.0, 328.0)	0.431
Presence of SRF	15/46	0 (0.0)	25 (54.4)	<0.001
Presence of IRC	15/46	4 (26.7)	17 (37.0)	0.466
Presence of RPED	15/46	5 (33.3)	27 (58.7)	0.088
EZ disruption	15/46	7 (46.7)	30 (65.2)	0.202
Presence of HRDs (inner)	15/46	2 (13.3)	9 (19.6)	0.716
Presence of HRDs (outer)	15/46	5 (33.3)	19 (41.3)	0.763
Presence of SHRM (μm)	15/46			0.144
0		10 (66.7)	19 (41.3)	
<500		4 (26.7)	14 (30.4)	
>500		1 (6.7)	13 (28.3)	
Choroidal thickness (μm)	15/46	200.0 (100.0, 250.0)	275.0 (225.0, 325.0)	0.002

Values are median (interquartile range) or sample size and proportion (%). Abbreviations: CMT, central macular thickness; EZ, ellipsoid zone; HRDs, hyperreflective dots; IRC, intraretinal cysts; MAR, the minimum angle of resolution; RPED, retinal pigment epithelium detachment; SHRM, subretinal hyperreflective material; and SRF, subretinal fluid.

**Table 3 ijms-24-07285-t003:** Changes in functional and anatomical outcomes after anti-VEGF treatment.

Variables	Eyes (OSA/Non-OSA)	OSA	Non-OSA	*p*-Value
At 1 month from baseline				
Delta changes in Log MAR	14/45	0.0 (−0.3, 0.3)	0.0 (−0.3, 0.1)	0.620
Delta changes in CMT	14/45	−103.5 (−141.0, −37.0)	−55.0 (−124.0, −10.0)	0.165
Changes in SRF	14/45			0.431
no difference		9 (64.3)	35 (77.8)	
increase		0 (0.0)	1 (2.2)	
decrease		5 (35.7)	9 (20.0)	
Changes in IRC	14/45			0.852
no difference		12 (85.7)	38 (84.4)	
increase		0 (0.0)	1 (2.2)	
decrease		2 (14.3)	6 (13.3)	
Changes in RPED	14/45			0.444
no difference		13 (92.9)	36 (80.0)	
increase		0 (0.0)	4 (8.9)	
decrease		1 (7.1)	5 (11.1)	
Changes in EZ disruption	14/45			0.364
no difference		13 (92.9)	35 (77.8)	
increase		1 (7.1)	5 (11.1)	
decrease		0 (0.0)	5 (11.1)	
Changes in HRDs (inner)	14/45			0.513
no difference		12 (85.7)	34 (75.6)	
increase		1 (7.1)	9 (20.0)	
decrease		1 (7.1)	2 (4.4)	
Changes in HRDs (outer)	14/45			0.658
no difference		6 (42.9)	25 (55.6)	
increase		3 (21.4)	9 (20.0)	
decrease		5 (35.7)	11 (24.4)	
Changes in SHRM	14/45			0.013
no difference		9 (64.3)	37 (82.2)	
increase		5 (35.7)	3 (6.7)	
decrease		0 (0.0)	5 (11.1)	
Delta changes in choroidal thickness	14/45	−25.0 (−50.0, 0.0)	−25.0 (−50.0, 50.0)	0.142
At 3 months from baseline				
Delta changes in Log MAR	15/46	−0.1 (−0.4, 0.0)	0.0 (−0.3, 0.2)	0.590
Delta changes in CMT	15/46	−51.0 (−221.0, −17.0)	−46.5 (−180.0, −8.0)	0.471
Changes in SRF	15/46			0.009
no difference		5 (33.3)	33 (71.7)	
increase		0 (0.0)	2 (4.4)	
decrease		10 (66.7)	11 (23.9)	
Changes in IRC	15/46			0.693
no difference		11 (73.3)	36 (78.3)	
increase		1 (6.7)	1 (2.2)	
decrease		3 (20.0)	9 (19.6)	
Changes in RPED	15/46			0.284
no difference		13 (86.7)	32 (69.6)	
increase		0 (0.0)	6 (13.0)	
decrease		2 (13.3)	8 (17.4)	
Changes in EZ disruption	15/46			0.221
no difference		13 (86.7)	36 (78.3)	
increase		2 (13.3)	3 (6.5)	
decrease		0 (0.0)	7 (15.2)	
Changes in HRDs (inner)	15/46			0.781
no difference		13 (86.7)	37 (80.4)	
increase		2 (13.3)	8 (17.4)	
decrease		0 (0.0)	1 (2.2)	
Changes in HRDs (outer)	15/46			0.769
no difference		10 (66.7)	27 (58.7)	
increase		2 (13.3)	10 (21.7)	
decrease		3 (20.0)	9 (19.6)	
Changes in SHRM	15/46			0.785
no difference		12 (80.0)	35 (76.1)	
increase		1 (6.7)	6 (13.0)	
decrease		2 (13.3)	5 (10.9)	
Delta changes in choroidal thickness	15/46	−25.0 (−50.0, 0.0)	−16.5 (−32.0, 12.5)	0.302

Values are median (interquartile range) or sample size and proportion (%). Abbreviations: CMT, central macular thickness; EZ, ellipsoid zone; HRDs, hyperreflective dots; IRC, intraretinal cysts; MAR, the minimum angle of resolution; RPED, retinal pigment epithelium detachment; SHRM, subretinal hyperreflective material; and SRF, subretinal fluid.

## Data Availability

The data presented in this study are available upon request from the corresponding author. The data are not publicly available due to ethical restrictions.

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
