# Peer review of "Functional and Anatomical Outcomes of Anti-Vascular Endothelial Growth Factor Treatment for Exudative Age-Related Macular Degeneration with or without Obstructive Sleep Apnea"

_ijms, 2023, doi:10.3390/ijms24087285_

Round 1

Reviewer 1 Report

I think the research topic is good, and the method for obtaining results is logical. However, we need to be careful when interpreting the results due to the small sample size. In my opinion, the essential data that must be presented are the severity of OSA at the time of diagnosis and whether the patients are actively undergoing treatment for OSA. This is because the response to Anti-VEGF may differ between patients who are actively treating OSA and those who are not. It is also necessary to show which type of Anti-VEGF was used in the study. Additionally, I believe the last paragraph in the Introduction section about OCT is unnecessary. Finally, it would be helpful to make Baseline, 1 month, and 3 months more distinguishable in Table 2."

Reviewer 2 Report

The researchers studied functional and anatomical outcomes of anti-vascular endothelial growth factor treatment for exudative age-related macular degeneration with or without obstructive sleep apnea. There are few previous works and it looks like an extension of that in 65 patients for which data were collected retrospectively, who received intravitreal injections (IVIs) of anti-VEGF drugs at Chang Gung Memorial Hospital, Chiayi . 

The observation and comment to improve the manuscripts are: 

1. No line number, so difficult to mention them in comments

2. Sentence on Page one: However, little research has investigated the treatment outcomes of patients with AMD and OSA is look bit controversial as following work available, 

Associations of sleep apnoea with glaucoma and age-related macular degeneration: an analysis in the United Kingdom Biobank and the Canadian Longitudinal Study on Aging | BMC Medicine | Full Text (biomedcentral.com)

UNTREATED OBSTRUCTIVE SLEEP APNEA HINDERS RESPONSE TO BEVACIZUMAB IN AGE-RELATED MACULAR DEGENERATION - PubMed (nih.gov)

The Associations of Obstructive Sleep Apnea and Eye Disorders: Potential Insights into Pathogenesis and Treatment | SpringerLink

"Association Between Age-Related Macular Degeneration and Sleep-Disorde" by Jeffrey A. Nau (waldenu.edu)

Authors must take such work into the note and elaborate in the introduction (Not limited to the above only) 

3. Page 2: Sentence: hough there have been reported that patients with exudative AMD and OSA have exhibited poor outcomes after anti–vascular endothelial growth factor (anti-VEGF) treatment [13,14], a general consensus has not been reached.

The author cited two pieces of literature here, however there few more works available, that should be discussed and then conclude in the sentence and if required modify as per the following work results (not limited to three only, search other similar works),

Poor responders to bevacizumab pharmacotherapy in age-related macular degeneration and in diabetic macular edema demonstrate increased risk for obstructive sleep apnea - PubMed (nih.gov)

Seven-year outcomes following intensive anti-vascular endothelial growth factor therapy in patients with exudative age-related macular degeneration | BMC Ophthalmology | Full Text (biomedcentral.com)

Visual Outcomes after Anti-VEGF Therapy for Exudative Age-Related Macular Degeneration in a Real-Life Setting - PubMed (nih.gov)

4. Please cite paragraphs, 2.1., 2.2, 2.3, and 2.4, baseline characteristics from previous studies, if not novel procedure or protocol. 

5. Table 1 results, is there any particular reseason, for more male patients than female (9 and 6). 

6. Fifteen Fifteen of the 65 patients, Make, Fifteen in number e.g.15
